# Elimination of Chirality in Three-Dimensionally Confined Open-Access Microcavities

**DOI:** 10.3390/nano13121868

**Published:** 2023-06-16

**Authors:** Yiming Li, Yuan Li, Xiaoxuan Luo, Chaowei Guo, Yuanbin Qin, Hongbing Fu, Yanpeng Zhang, Feng Yun, Qing Liao, Feng Li

**Affiliations:** 1Key Laboratory for Physical Electronics and Devices of the Ministry of Education & Shaanxi Key Lab of Information Photonic Technique, School of Electronic Science and Engineering, Faculty of Electronic and Information Engineering, Xi’an Jiaotong University, Xi’an 710049, China; yue226@stu.xjtu.edu.cn (Y.L.); lxx404@stu.xjtu.edu.cn (X.L.); ypzhang@mail.xjtu.edu.cn (Y.Z.); fyun2010@mail.xjtu.edu.cn (F.Y.); 2Beijing Key Laboratory for Optical Materials and Photonic Devices, Department of Chemistry, Capital Normal University, Beijing 100048, China; 2210601006@cnu.edu.cn (Y.L.); 6562@cnu.edu.cn (H.F.); 3Center for Advancing Materials Performance from the Nanoscale (CAMP-Nano), State Key Laboratory for Mechanical Behavior of Materials, Xi’an Jiaotong University, Xi’an 710049, China; cwguo@xjtu.edu.cn (C.G.); qinobin@xjtu.edu.cn (Y.Q.); 4Solid-State Lighting Engineering Research Center, Xi’an Jiaotong University, Xi’an 710049, China

**Keywords:** organic microcrystal, spin-orbit coupling, microcavity

## Abstract

The emergent optical activity (OA) caused by anisotropic light emitter in microcavities is an important physical mechanism discovered recently, which leads to Rashba–Dresselhaus photonic spin-orbit (SO) coupling. In this study, we report a sharp contrast of the roles of the emergent OA in free and confined cavity photons, by observing the optical chirality in a planar–planar microcavity and its elimination in a concave–planar microcavity, evidenced by polarization-resolved white-light spectroscopy, which agrees well with the theoretical predictions based on the degenerate perturbation theory. Moreover, we theoretically predict that a slight phase gradient in real space can partially restore the effect of the emergent OA in confined cavity photons. The results are significant additions to the field of cavity spinoptronics and provide a novel method for manipulating photonic SO coupling in confined optical systems.

## 1. Introduction

Photonic spin-orbit (SO) coupling is a widely studied area in free optics [1,2,3], dielectric [4,5,6], and surface plasmonic systems [7,8,9,10,11,12]. As one of the most promising on-chip platforms for manipulation of SO interaction, the Fabry–Perot (F–P) microcavities exhibit rich physical properties of polariton dynamics under the effective photonic gauge field associated with various polarization splittings [13], leading to potential applications in all-optically controlled on-chip devices [14,15]. Particularly, the transverse electric-transverse magnetic (TE-TM) splitting has led to important phenomena, such as optical spin-Hall effect [16,17], dark-half solitons [18], and spin vortices [19] in polariton quantum fluids. Recently, other mechanisms generating the effective gauge field are widely reported in addition to the TE-TM splitting, by combining the effects of anisotropic materials and external magnetic field. The anisotropy leads to the effect of emergent optical activity (OA) in preserving the time reversal symmetry (TRS) [20], which results in the chirality of light propagation carrying an asymmetric circular photon spin. This has allowed for the demonstration of Rashba–Dresselhaus SO coupling [21,22], non-trivial topological bands and valleys [20,23,24,25], Voigt exceptional points [26], divergent quantum metric [27], and helical polariton lasing [25,28].

In addition to the SO coupling of the two-dimensional (2D) free photonic gas and polariton fluid which show spectra of continuum with the variation of wavevector, the investigations of the SO coupling effects on totally confined photonic systems have led to significant results, including the eigenstates of spin vortices in concave–planar open-access microcavities [29] and Benzene-like photonic molecules [30], which would be used as low threshold tunable vector-vortex lasers. These phenomena result from the coupling between the confined orbital angular momentum (OAM) and the spin angular momentum (SAM) of the cavity photons with isotropic active materials, in which the photonic effective field is solely determined by the TE-TM splitting. On the other hand, the role of the anisotropic active media remains unexplored, especially for the tightly confined mode, which is a standing wavepacket containing a large range of symmetrically distributed wavevectors. It would be interesting to investigate the spin property of the confined modes under the influence of optical chirality restricted to TRS.

In this article, we perform theoretical and experimental studies on the polarization properties of confined modes in a concave–planar open-access microcavity embedded with anisotropic active materials. Theoretically, we find that the optical chirality induced by the emergent OA is completely eliminated by the confinement potential, which is evidenced by the experimental studies with polarization-resolved microscopy of white-light reflectivity. Moreover, we predict that the effect of the emergent OA can be partially restored if a spatial phase gradient can be introduced. The results provide advanced theoretical frame and experimental platform for manipulating photonic SO coupling in confined optical systems.

## 2. Materials and Methods

The experimental system includes an open-access microcavity embedded with anisotropic organic microcrystals, as illustrated in Figure 1a. The open-access microcavity consists of a bottom-distributed Bragg reflector (DBR) of nine pairs of Ta_2_O_5_/SiO_2_ deposited on a silica substrate and a top DBR of the same structure [Figure 1b], which is etched into concave shapes by a focused ion beam [31]. These concave shapes constitute a two-dimensional array containing four rows of eight concave shapes with radii of curvatures (ROC) of 50 μm, 20 μm, 12 μm, and 7 μm, which exhibit mirror sizes (in diameter) of 16 μm, 10 μm, 8 μm, and 6 μm, respectively. The adjacent concave shapes are separated by a distance of 25 μm uniformly, indicating a total size of the arrays of around 200 μm by 100 μm. The DBRs are designed to have a center wavelength of 490 nm and a high-reflectivity bandwidth of ~108 nm. The two DBRs are adjusted with nanopositioners to provide fine detuning of cavity length and spatial position of the concave mirror. In this study, the distance between the top and bottom DBRs is tuned around 10 μm. In particular, the top mirror contains both concave and planar regions, making the system a concave–planar (resp. planar–planar) microcavity when the concave (resp. planar) part is positioned under the excitation light beam. The organic microcrystal of 2,2-difluoro-4,6-bis(4-methoxyphenyl)-2H-1l3,3,2l4-dioxaborinine (BF2MO2), shown in Figure 1c,d, is fabricated and transferred onto the bottom DBR to serve as the active media. The optical property of the organic microcavity is characterized by polarization-resolved micro-reflectivity measurement with white-light excitation.

Whilst the details of the DBR fabrication and the micro-reflectivity setup are included in the Appendix A and Appendix B, we describe in detail the fabrication process of the organic microcrystal of BF2MO2. First, the small organic molecule BF2MO2 was synthesized via a single-step reaction of boron trifluoride (BF_3_·OEt_2_) boronation in dichloromethane (CH_2_Cl_2_). A mixture of 1,3-Bis(4-methoxyphenyl)-1,3-propanedione and boron trifluoride diethyl etherate was added into a flask equipped with a reflux condenser and heated in an oil bath at 60 °C. After silica gel separation, BF2MO2 was obtained in 85% yield. The microstructures of BF2MO2 were obtained by a facile micro-spacing in-air sublimation method. Briefly, a piece of glass, as the bottom substrate on which our synthetic compounds were dispersed, was placed on a hot stage. The glass substrate with DBR, as the top substrate, was placed on the bottom substrate with a spacing distance of 170 μm separated by two tiny pieces of glass with rectangular shapes, such as railway sleepers. The growth procedure was implemented by heating the bottom substrate to sublime the materials and form microcrystals on the down surface of the top substrate. Micro-meter-scale BF2MO2 crystals grew uniformly on the down surface of the top substrate at a temperature of 220 °C. The obtained BF2MO2 microcrystals have regular morphologies with the length of 50–70 μm, the width of 20–50 μm, and the thickness of 450–650 nm.

## 3. The Behavior of Free Cavity Photons

Similar to the previously reported anisotropic materials, such as liquid crystal [22], Perylene [20,32], and DPAVBi [27], the BF2MO2 microcrystal exhibits a tilted optical axis with regard to the cavity mirrors, leading to the emergent optical activity which is well-known from the recent literature [20,27]. The system is described by the Rashba–Dresselhaus Hamiltonian, written in the circular polarization bases:(1)$$\left(\begin{array}{cc} \frac{\hbar^{2}k^{2}}{2m}+V+\zeta k_{x} & \beta_{0}+\beta k^{2}e^{-2i\varphi }\\ \beta_{0}+\beta k^{2}e^{2i\varphi } & \frac{\hbar^{2}k^{2}}{2m}+V-\zeta k_{x} \end{array}\right)$$
where *m* is the photonic effective mass, *k* is the in-plane momentum of cavity photons with *k_x_* being its component along *x* direction, and φ is the azimuthal angle. *ζ*, *β*, and *β*_0_ are the SO coupling coefficients associated with the emergent OA, the TE-TM splitting, and the linear birefringence splitting. *V* is the photonic potential defined by the lateral confinement, which is zero for planar–planar microcavities. At *V* = 0, the cavity photons behave similarly to the two-dimensional (2D) free particles, in which the emergent OA leads to Rashba–Dresselhaus-like dispersions [22] and gapped Dirac cones [20] when the two orthogonally polarized modes of opposite parity coalesce at zero and non-zero momenta, respectively. We construct the planar–planar microcavity by using the planar part of the top mirror, and measure the angular-resolved white-light reflectivity as shown in Figure 2a, which displays a clear feature of the Rashba–Dresselhaus dispersion, as previously demonstrated in liquid crystal microcavities [22]. In contrast to the liquid crystal cavity, which allows for tuning of the photonic energies of the ordinary light (o-light) and extraordinary light (e-light) modes by applying voltage, the energy detuning between the two modes at *k* = 0 (referred to as the o-e detuning hereafter) are not adjustable in our organic cavity, as the axis of the MO2BF2 crystal is fixed. Therefore, to achieve the desired o-e detuning, a large number of samples with different BF2MO2 thicknesses within a large range of wavelengths are measured for each sample. Indeed, Figure 2 is obtained from a sample with a preferred o-e detuning. It shows a number of mode groups in the measured wavelength range, in which we labeled four as groups I to IV. From the spectrum, we can see that group II is at zero o-e detuning, while the other groups are at non-zero o-e detuning. While the detuning of group I is very small, in comparison with the mode linewidths, groups III and IV display very clear energy splittings between the o and e lights. By calculating the energy-momentum relation of group II (*ζ* = 2*α* = 4*E*_0_/*k*_0_) [33,34], an OA coefficient *ζ* = 6 meV·μm is obtained.

The polarization of the mode groups is characterized by Stokes parameters, which can be derived from the polarization-resolved measurements, with *S*_1_, *S*_2_, *S*_3_ representing the horizontal-vertical, diagonal, and circular degrees of polarization, respectively (see Appendix B for details). Herein, the polarizations of the o and e lights are defined as the horizontal and vertical directions associated with *S*_1_. As the reflectivity of the DBRs decreases from group I to group IV, we observe shallower reflectivity dips (thereby higher white-light background) as the *Q* factor increases, due to the increased ratio between the intrinsic photonic disorder (induced by the sample surface inhomogeneity and the mechanical vibration of mirrors) and the mode linewidth. The white-light background leads to the fact that the total polarization degree $S_{0}=\sqrt{S_{1}^{2}+S_{2}^{2}+S_{3}^{2}}$ is significantly less than 1, which is smaller when the modes approach a shorter wavelength. This raises difficulties in reaching a decent comparison of the polarization properties between different mode groups, as well as the comparison between the experimental data with the simulation from Equation (1), in which the white-light background is not considered. To solve this problem, we extract points of each mode in the same angle (12°) in Figure 2b–d as marked, and then calculate the normalized Strokes parameters by $S_{i}^{\prime }=S_{i}/S_{0}$ (*i* = 1,2,3), which are shown as black squares in Figure 2e,f. It is clearly seen that the circular polarization degree drops, while the linear polarization degree increases with increasing o-e detuning in Figure 2e,f, as expected theoretically. The curves in Figure 2e,f show the calculated *S*_1_ and *S*_3_ at the angle of 12° (*k_x_* = 2 μm^−1^), as a function of the o-e detuning by varying the linear splitting term *β*_0_ in Equation (1), which fits well with the experimentally-derived values of $S_{1}^{\prime }$ and $S_{3}^{\prime }$. It is seen that even when the o and e modes are largely detuned as for groups III and IV, the circular polarization degree is still large, which is comparable with the linear one. We note that the angle of 12° is chosen for the plot as it approximately corresponds to the positions of energy minima of the dispersion curve, while other angles, if not very small (close to 0°) or very large (e.g., >45° in absolute value), generally show the same feature.

## 4. The Behavior of the Confined Cavity Photons: Theory

The situation with the confinement of a photonic potential, i.e., *V* ≠ 0, is, nevertheless, very different. The cavity photons become three-dimensionally (3D) confined, and thereby show discrete eigenenergies without the feature of dispersion. As these confined modes exhibit finite-sized spatial distributions whose Fourier transform leads to a finite range of momentum distributions, one needs to investigate whether the OA term $\pm \zeta k_{x}$ in Equation (1) has corresponding effects on the polarization of the modes. First, we consider a circular harmonic potential expressed by $V=\frac{1}{2}m\omega^{2}\left(x^{2}+y^{2}\right)$, with *ω* characterizing the depth of the potential. When the potential is significantly larger than the SO coupling terms, which have been demonstrated as true in the planar–concave open-access microcavities [29], the degenerated perturbation theory of standard quantum mechanics can be applied to Equation (1) to obtain the eigenvalues and eigenstates. Namely, the potential *V* is regarded as the unperturbed Hamiltonian and all the SO coupling terms, including *ζ*, *β,* and *β*_0_, are the perturbation terms. The Hamiltonian Equation (1) can be written as:(2)$$\hat{H}={\hat{H}}_{0}+{\hat{H}}^{\prime }=\left(\begin{array}{cc} \frac{\hbar^{2}k^{2}}{2m}+V & 0\\ 0 & \frac{\hbar^{2}k^{2}}{2m}+V \end{array}\right)+\left(\begin{array}{cc} \zeta k_{x} & \beta_{0}+\beta k^{2}e^{-2i\varphi }\\ \beta_{0}+\beta k^{2}e^{2i\varphi } & -\zeta k_{x} \end{array}\right),$$
where ${\hat{H}}_{0}$ and ${\hat{H}}^{\prime }$ are the unperturbed and permutation Hamiltonians. Under the circular harmonic potential *V* expressed above, the ground eigenstates of the unperturbed Hamiltonian ${\hat{H}}_{0}$ have the following form:(3)$$\psi_{1}=\psi_{0}\left(\begin{array}{c} 1\\ 0 \end{array}\right),$$
(4)$$\psi_{2}=\psi_{0}\left(\begin{array}{c} 0\\ 1 \end{array}\right),$$
where $\psi_{0}=\frac{1}{\sqrt{\pi }\sigma }e^{-\frac{x^{2}+y^{2}}{2\sigma^{2}}}$ ($\sigma =\sqrt{\hbar /m\omega }$) is the Gaussian-like field distribution constituting the ground state of the circular potential, and (1 0)*^T^* and (0 1)*^T^* represent the right (*σ*^+^) and left (*σ*^−^) circular polarizations, respectively. According to the principle of quantum mechanics, the elements of the perturbation matrix are derived by [35]:(5)$$H^{\prime }{}_{mn}=\left(\psi_{m},{\hat{H}}^{\prime }\psi_{n}\right),$$
where *m*, *n* are integers that fit in the size of the system dimension. By applying Equations (2) to (5), we derive the perturbation matrix:(6)$$H^{\prime }=\left(\begin{array}{cc} H_{11}^{\prime } & H_{12}^{\prime }\\ H_{21}^{\prime } & H_{22}^{\prime } \end{array}\right)=\left(\begin{array}{cc} 0 & \beta_{0}\\ \beta_{0} & 0 \end{array}\right),$$

Herein, the algorithm $k_{x}\to -i\frac{\partial }{\partial x}$, $k_{y}\to -i\frac{\partial }{\partial y}$ is applied. As a result, we can identify that the OA terms, characterized by ζ, completely disappear, indicating that the emergent OA has no effect on the confined photons. Meanwhile, the TE-TM splitting term *β* does not play any role. According to quantum mechanics, the eigenvalues and eigenvectors of *H*′ are the first-order eigenenergy corrections and the zeroth-order eigenfunctions of the system. Therefore, the calculated eigenenergies and eigenstates are:(7)$$E_{1}=\beta_{0},$$
(8)$$E_{2}=-\beta_{0},$$
(9)$$\Psi_{1}=\frac{\psi_{0}}{\sqrt{2}}\left(\begin{array}{c} 1\\ 1 \end{array}\right),$$
(10)$$\Psi_{2}=\frac{\psi_{0}}{\sqrt{2}}\left(\begin{array}{c} -1\\ 1 \end{array}\right),$$
indicating that the modes at non-zero o-e splitting ($\beta_{0}\;\neq \;0)$ are in pure linear polarization. At zero detuning of ($\beta_{0}=0)$, the modes of different polarizations degenerate, leaving the polarization of the mode uncertain, which might be subject to effects of symmetry breaking induced by the photonic system.

We further study higher-order transverse modes of the confined systems, which are typically Lagurre–Gaussian (LG) or Hermite–Gaussian (HG) modes. It has been shown that the modes of the first excited manifold are strongly affected by the TE-TM splitting term *β* and form spin vortices and skyrmions [29]. To know whether these modes also interact with ζ, we apply the perturbation theory using the HG modes as the unperturbed eigenstates:(11)$$\psi_{1}=\frac{xe^{-\frac{x^{2}+y^{2}}{2\sigma^{2}}}}{\sqrt{\pi /2}\sigma^{2}}\left(\begin{array}{c} 1\\ 0 \end{array}\right),$$
(12)$$\psi_{2}=\frac{xe^{-\frac{x^{2}+y^{2}}{2\sigma^{2}}}}{\sqrt{\pi /2}\sigma^{2}}\left(\begin{array}{c} 0\\ 1 \end{array}\right),$$
(13)$$\psi_{3}=\frac{ye^{-\frac{x^{2}+y^{2}}{2\sigma^{2}}}}{\sqrt{\pi /2}\sigma^{2}}\left(\begin{array}{c} 1\\ 0 \end{array}\right),$$
(14)$$\psi_{4}=\frac{ye^{-\frac{x^{2}+y^{2}}{2\sigma^{2}}}}{\sqrt{\pi /2}\sigma^{2}}\left(\begin{array}{c} 0\\ 1 \end{array}\right),$$
which yields the 4 × 4 perturbation matrix, calculated using Equations (2) and (5):(15)$$H^{\prime }=\left(\begin{array}{cccc} 0 & -\frac{\beta }{\sigma^{2}}+\beta_{0} & 0 & \frac{\beta }{\sigma^{2}}\\ -\frac{\beta }{\sigma^{2}}+\beta_{0} & 0 & -\frac{i\beta }{\sigma^{2}} & 0\\ 0 & \frac{i\beta }{\sigma^{2}} & 0 & \frac{\beta }{\sigma^{2}}+\beta_{0}\\ -\frac{i\beta }{\sigma^{2}} & 0 & \frac{\beta }{\sigma^{2}}+\beta_{0} & 0 \end{array}\right),$$
in which no matrix element contains ζ, as all the integrals containing ζ result in a value of zero. Therefore, the first excited manifold, though strongly affected by the TE-TM splitting, is not affected by the effect of emergent OA, and thereby shows the same features as reported in [29].

The absence of chirality in the confined mode is physically related to the time-reversal symmetry of the emergent OA, which changes its sign with $k_{x}$, therefore, the effect cancels if the system is symmetric around $k_{x}=0$. To break the symmetry along $k_{x}$, we consider the situation that the eigenstate of the system exhibits a phase gradient *η*, namely:(16)$$\psi^{\prime }{}_{0}=\psi_{0}e^{i\eta x},$$

The coefficient ζ will be included in the perturbation matrix of SO coupling:(17)$$H^{\prime }=\left(\begin{array}{cc} \zeta \eta & \beta_{0}+\beta \eta^{2}\\ \beta_{0}+\beta \eta^{2} & -\zeta \eta \end{array}\right),$$
and yields the eigenenergies and eigenstates:(18)$$E_{1}=-a,$$
(19)$$E_{2}=a,$$
(20)$$\Psi_{1}=\frac{1}{\sqrt{1+\frac{{\left(-\zeta \eta +a\right)}^{2}}{{\left(\beta_{0}+\beta \eta^{2}\right)}^{2}}}}\left(\begin{array}{c} -\frac{-\zeta \eta +a}{\beta_{0}+\beta \eta^{2}}\\ 1 \end{array}\right),$$
(21)$$\Psi_{2}=\frac{1}{\sqrt{1+\frac{{\left(-\zeta \eta -a\right)}^{2}}{{\left(\beta_{0}+\beta \eta^{2}\right)}^{2}}}}\left(\begin{array}{c} -\frac{-\zeta \eta -a}{\beta_{0}+\beta \eta^{2}}\\ 1 \end{array}\right),$$
in which:(22)$$a=\sqrt{\beta_{0}^{2}+2\beta \beta_{0}\eta^{2}+\zeta^{2}\eta^{2}+\beta^{2}\eta^{4}}$$

In this situation, the eigenstates split in energy and show non-zero circular polarization degree depending on the quantitative relation between *ζ*, *β*, and *β*_0_. In general, as seen from Equations (20) and (21), $\Psi_{1}$ and $\Psi_{2}$ show neither identical nor antisymmetric circular polarization degrees, which is the typical consequence of symmetry breaking. In particular, when *η* = 0, the eigenstates reduce to the purely linearly polarized ones as discussed above.

The physical meaning of this phase gradient can be associated with the inhomogeneity of the active layer. The gradient of the thickness of the active layer would result in a spatial variation of the photonic potential inside the cavity, leading the cavity photons to travel along *x*, which corresponds to the term *e^iηx^* in Equation (16). Since a photonic potential is generally significantly smaller than the confinement potential *V*, the cavity photons are not able to escape from the cavity along *x* during the cavity lifetime. Therefore, the modes are still well-confined even with the lateral phase gradient. Nevertheless, if *η* is sufficiently large to be comparable with *V*, the cavity no longer confines the photons, which is beyond the scope of the investigation in this manuscript.

## 5. The Behavior of the Confined Cavity Photons: Experiment

The experimental system supporting non-zero potential *V* is the concave−planar microcavity, in which the circular concave mirror is approximately regarded as a circular harmonic potential for photons [29]. The concave mirror has a ROC of ~20 μm and a diameter of ~10 μm. The incident white-light beam is focused on the concave mirror by a 100× objective lens. The polarization-resolved reflectivity spectra of the discrete laterally confined modes are shown in Figure 3, presenting the measurements at horizontal (0°), vertical (90°), diagonal (45°), anti-diagonal (−45°) linear polarizations, as well as $\sigma^{+}$ and $\sigma^{-}$ circular polarizations. The left and right vertical axes in Figure 3c,e,g,i correspond to the calculated Stokes parameter *S_i_* and the normalized Stokes parameter *S_i_*′, respectively. The visibility is the best for the ground lateral mode LG_00_, which has the best spatial matching with the focused excitation light beam, while some higher-order lateral modes (mostly the LG_01_ mode) still appear as shallower dips compared with the ground mode in the same longitudinal group. For simplicity, we continue to analyze the LG_00_ mode, wherein we add rectangular-shaped light red and light purple backgrounds for clarity. Only the light red backgrounds are applied in the graphs with small o-e detuning, i.e., Figure 3b–e, in which the o-e mode splittings are smaller than the mode linewidths. As expected, the mode groups I–IV appear in the spectra, showing the same o-e detuning as in the planar–planar microcavity characterized by Figure 2, since the birefringence layer is unchanged. We first focus on mode group II in Figure 3d, which shows zero o-e detuning. In fact, as seen from the spectra of 0° and 90° polarizations, the two modes exhibit a very small splitting between o-e, which is nevertheless completely negligible compared with the mode linewidth in order that it can be considered as $\beta_{0}=0$. As the theoretical calculation predicts no preference of polarization in this situation, the degenerate mode displays $S_{1}^{\prime }=-0.4$, $S_{2}^{\prime }=-0.35$, $S_{3}^{\prime }=-0.5$ (at the wavelength of 547.9 nm) in the measurement in Figure 3e, corresponding to a state of elliptical polarization possibly determined by the structural symmetry breaking, such as defect-caused phase pinning. Group I in Figure 3b shows a non-negligible splitting of the o and e modes, which is comparable with the mode linewidth, when measured in the 0° and 90° polarizations. The spectra overlap between the two modes makes them indistinguishable in wavelength in the overall, diagonal, and circular polarization basis, and thus cannot be analyzed by our theoretical model. On the other hand, the o-e splitting in mode groups III and IV [Figure 3f,h] is clearly larger than the mode linewidth in both the overall and polarization-resolved spectra, providing sufficient clarity for analysis. Indeed, the two modes are almost horizontally and vertically polarized, with very small diagonal and circular polarizations [$S_{1}^{\prime }$, $S_{2}^{\prime }$, and $S_{3}^{\prime }$ shown in Figure 3g,i], indicating that the elimination of the effect of the OA term is associated with ζ, as predicted theoretically.

Finally, we note that parts of the theoretical predictions have not yet been proven experimentally due to the limitation of the experimental platform. First, it is difficult to measure and resolve higher-order transverse modes with white-light reflectivity in our experimental system. In addition, fabricating samples with designed spatial or phase asymmetry is difficult. In particular, it is likely that the small non-zero circular polarization degrees in Figure 3 (group III and even group IV) originate from the phase asymmetry induced possibly by the active layer inhomogeneity, while the details of the relevant structural parameters are, however, difficult to obtain. Therefore, we consider these investigations out of the scope of this paper and a subject of future research.

## 6. Conclusions

In conclusion, we have theoretically proven the loss of role of the emergent OA in the fully-confined cavity photons, which is nevertheless a striking effect on the 2D free cavity photons. We experimentally demonstrated the elimination of chirality using a tunable concave–planar microcavity system. Our results highlighted an important physical mechanism for understanding the role of the emergent OA in optical microcavities, and added considerable knowledge to the cavity spinoptronics in confined systems. Furthermore, we can expect that the polarization of the confined LG modes can be tuned by an interplay between the emergent OA and the structural symmetry. In particular, when the confinement potential becomes shallow to be comparable with the emergent OA where the perturbation theory no longer holds, the polarization pattern will be very likely to display a competition between the confined and free cavity photons. All these mechanisms are interesting for further investigations, which may lead to novel designs for polarization-based photonic devices.

## Figures and Tables

**Figure 1 nanomaterials-13-01868-f001:**
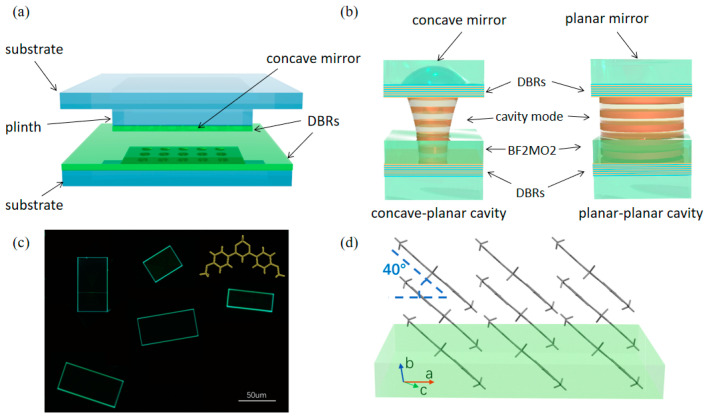
Schematic of the experimental system. (**a**) Illustrative picture of the open-access microcavity fabricated on silica substrates. (**b**) An enlarged view of the concave–planar configuration (left) and planar–planar configuration (right). (**c**) BF2MO2 molecular structure and the microscopic image of the microcrystal. (**d**) The molecular arrangement in the microcrystal.

**Figure 2 nanomaterials-13-01868-f002:**
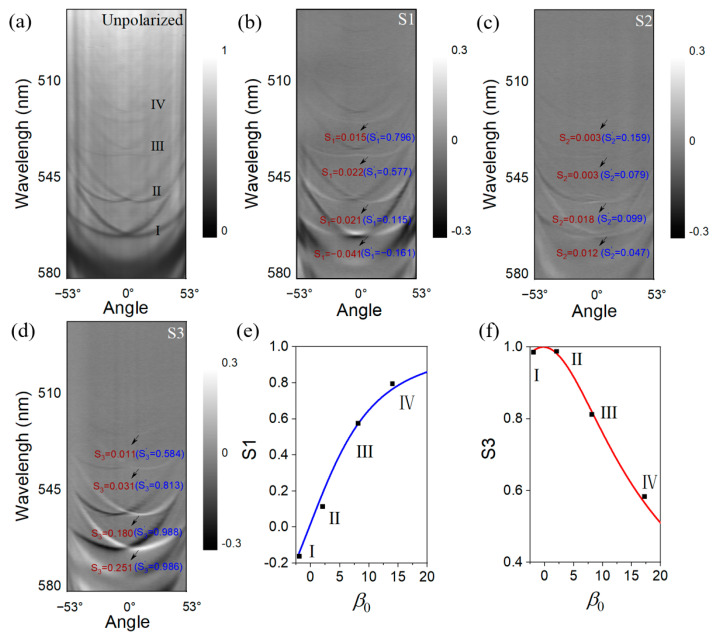
Characterization of the planar–planar microcavity. (**a**–**d**) Angular-resolved reflectivity spectrum of the planar–planar organic microcavity, including measured spectra without polarization (**a**), and displayed by the Stokes parameters of *S*_1_ (**b**), *S*_2_ (**c**), and *S*_3_ (**d**). Both the original Stokes parameters $S_{i}$ and the normalized ones $S_{i}^{\prime }$ are labeled in the graphs. (**e**,**f**) The simulated *S*_1_ (**e**) and *S*_3_ (**f**) at the angle of 12° as a function of o-e detuning. The experimental data of $S_{1}^{\prime }$ and $S_{3}^{\prime }$ are plotted as black squares.

**Figure 3 nanomaterials-13-01868-f003:**
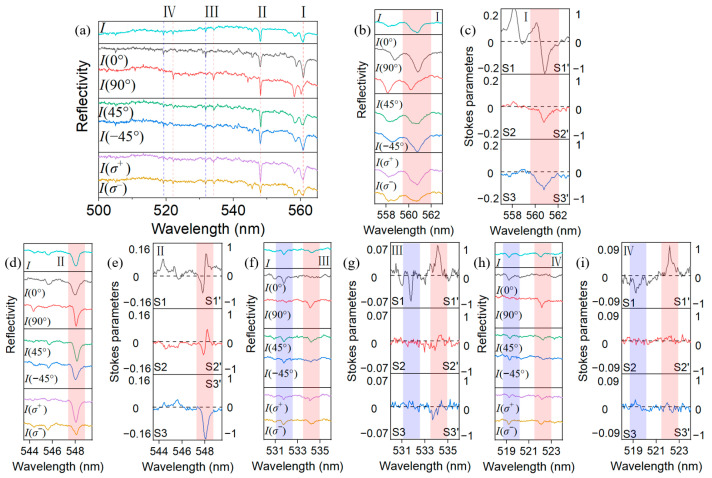
The polarization-resolved white-light reflectivity spectrum of a concave–planar microcavity embedded with a BF2MO2 thin film. The whole spectral range is presented in (**a**) and the two modes in the sample group are labeled by blue dashes and red dotted lines. Enlarged wavelength ranges for different mode groups I–IV are presented in (**b**,**d**,**f**,**h**). Meanwhile, (**c**,**e**,**g**,**i**) are the calculated polarization degrees for comparison. In each graph, the left and right vertical axes represent the values of $S_{i}$ and $S_{i}^{\prime }$, respectively. For clarity, the LG_00_ modes are covered by red and purple light-color box background. In situations where the o-e detuning is very small to separate the two orthogonally polarized modes, namely, in graphs (**b**–**e**), only a red box is applied.

## Data Availability

The data presented in this study are available on request from the corresponding author.

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
