# Peer review of "Elimination of Chirality in Three-Dimensionally Confined Open-Access Microcavities"

_nanomaterials, 2023, doi:10.3390/nano13121868_

Round 1

Reviewer 1 Report

1st review of "Elimination of chirality in three-dimensionally confined open-access microcavities" by Y. Li et al.

The authors study open-acces microcavities in 3D which lead to elimination of the chirality. Presented study are timly and interesting. Before acceptance I have few suggestion.

IMO on fig. 1 should be added some symbols (or descriptions) helping read it.

Fig. 2 and 3 can be bigger - now labels are not too well visible.

technical remarks:

- caption and fig. should be on the same page

Reviewer 2 Report

Comments in the appended file.

Other observations on the language:

·          At line 38, what does “reaches beyond” mean?

·          At line 76, planar-planar and not planar-planEr.

·          At line 98, what are glass sleepers?

·          At line 196 “which might BE subject”.

·          At lines 228-229, try to reformulate the expression “which is then not the situation we discuss”.

·          At line 289 chirality and not charily.

·          At line 290, “Our results highlight an important physical mechanism” or you can use the verb describes, illustrates etc.

I am not an English native speaker, but I suggest a careful check of the language.
